# Cardiac RGS Proteins in Human Heart Failure and Atrial Fibrillation: Focus on RGS4

**DOI:** 10.3390/ijms24076136

**Published:** 2023-03-24

**Authors:** Jordana I. Borges, Malka S. Suster, Anastasios Lymperopoulos

**Affiliations:** Laboratory for the Study of Neurohormonal Control of the Circulation, Department of Pharmaceutical Sciences, Barry and Judy Silverrman College of Pharmacy, Nova Southeastern University, Fort Lauderdale, FL 33328-2018, USA

**Keywords:** atrial fibrillation, cardiac myocyte, cyclic AMP, G protein-coupled receptor, heart failure, regulator of G protein signaling-4, signal transduction

## Abstract

The regulator of G protein signaling (RGS) proteins are crucial for the termination of G protein signals elicited by G protein-coupled receptors (GPCRs). This superfamily of cell membrane receptors, by far the largest and most versatile in mammals, including humans, play pivotal roles in the regulation of cardiac function and homeostasis. Perturbations in both the activation and termination of their G protein-mediated signaling underlie numerous heart pathologies, including heart failure (HF) and atrial fibrillation (AFib). Therefore, RGS proteins play important roles in the pathophysiology of these two devasting cardiac diseases, and several of them could be targeted therapeutically. Although close to 40 human RGS proteins have been identified, each RGS protein seems to interact only with a specific set of G protein subunits and GPCR types/subtypes in any given tissue or cell type. Numerous in vitro and in vivo studies in animal models, and also in diseased human heart tissue obtained from transplantations or tissue banks, have provided substantial evidence of the roles various cardiomyocyte RGS proteins play in cardiac normal homeostasis as well as pathophysiology. One RGS protein in particular, RGS4, has been reported in what are now decades-old studies to be selectively upregulated in human HF. It has also been implicated in protection against AFib via knockout mice studies. This review summarizes the current understanding of the functional roles of cardiac RGS proteins and their implications for the treatment of HF and AFib, with a specific focus on RGS4 for the aforementioned reasons but also because it can be targeted successfully with small organic molecule inhibitors.

## 1. Introduction

G protein-coupled receptors (GPCRs) are the single largest class of pharmaceutical targets, with over 35% of the total drugs currently used in clinical practice being ligands (i.e., binding directly) of these receptors at the cell membrane [1]. GPCRs are crucial regulators of almost every cellular physiological process, such as vision, taste and smell perception, neurotransmission, metabolism, blood coagulation, cell growth and death, cardiac function, vascular reactivity, and blood pressure [2]. This is largely because they always reside at the plasma membrane, thereby mediating the signal from the vast majority of extracellular stimuli that cannot pass across the cell membrane (e.g., molecules that are ionized or not lipophilic enough). It follows that impairments in GPCR function or functional numbers (receptor density) leads to abnormal signaling, activity, or ligand binding properties of the receptor, which, in turn, results in various pathologies, depending on the physiological process that is normally regulated by the dysfunctional receptor. If the GPCR in question is expressed in the cardiovascular system, then cardiovascular pathologies ensue, such as heart failure (HF), dilated cardiomyopathy, cardiac hypertrophy, hypertension, atherosclerosis, blood clotting, angina, etc. [3,4]. All (class A) GPCRs share a common core motif of seven largely hydrophobic α helices, each spanning the entire plasma membrane (seven transmembrane (TM)-spanning or heptahelical receptors) [5]. The TM1, TM2, and TM4 helices have largely structural roles, i.e., do not participate directly in agonist binding (on the extracellular side) or G protein activation (on the cytoplasmic side), as the other four TM helices (TM3, TM5, TM6, and TM7) do [6]. Nevertheless, all seven TM helices are essential for the receptor’s heptahelical motif, which, in turn, is essential for the formation, on its cytoplasmic face, of a pocket between intracellular loop (ICL)-2 and the C-terminal tail (helix H8) of the receptor, i.e., between the cytoplasmic ends of helices TM3 and TM7. When the GPCR is inactive, this pocket is sterically blocked for any interaction with G proteins by the presence of ICL3, formed by the intracellular ends of TM5 and TM6, protruding into the space between the cytoplasmic ends of TM3 and TM7 and occupying it [7]. Upon agonist binding on the extracellular side of the receptor, this pocket opens up to a significant extent, thanks, mainly, to an outward movement of the cytoplasmic half of TM6 away from TM3 and TM7 (and closer to TM5), in addition to other conformational changes [8]. In this conformational state of the receptor, the Gα subunit of the heterotrimeric G protein can now dock onto the receptor, with its C-terminal amphipathic and extremely dynamic α5 helix probing and interacting with hydrophobic residues deep inside the 7TM helix core [9,10]. This receptor-α5 helix interaction leads to profound conformational rearrangements throughout the Gα subunit, which result in ejecting guanosine diphosphate (GDP) and movements of Gα’s Ras and alpha-helical (AHD) domains away from each other to make room for binding guanosine triphosphate (GTP), much more abundant than GDP in the cytoplasm, instead [7,10,11]. The separation of Ras and AHD domains away from each other is necessary for GTP binding because GDP is “buried” between them when bound to Gα and shielded from the cytoplasmic aqueous environment, from which GTP has to emerge to bind to the nucleotide-free Gα subunit [12]. In any case, the binding of GTP “locks” the Gα subunit in a conformation that can no longer accommodate the Gβγ dimer (nor the receptor) [13,14], and thus, GTP-bound Gα, like the free Gβγ dimer, is now free to interact with effectors and start signaling, i.e., is active. This occurs mainly because the switch II region of Gα’s Ras domain, which forms the interface with Gβ in the assembled heterotrimer, now makes contacts with the γ-phosphate of GTP, instead [12,13]. Therefore, GPCRs act as guanine nucleotide exchange factors (GEFs) for the heterotrimeric G proteins, in essence “freeing” Gα subunits from Gβγ-“inhibition” to activate or inhibit effectors (e.g., enzymes and ion channels), eliciting a variety of cellular responses. The regulation of the duration of a GPCR signal is of paramount importance for cellular homeostasis. Therefore, the cell utilizes various ways to terminate the GPCR signal, starting with two major processes at the level of the cell membrane. One of them operates on the receptor itself and involves GPCR phosphorylation by GPCR-kinases (GRKs), followed by arrestin binding (homologous or agonist-dependent receptor desensitization) [15,16]. Second messenger-dependent kinases, such as protein kinase A (PKA) and protein kinase C (PKC), can also phosphorylate the receptor, terminating or even preventing signaling (heterologous or agonist-independent receptor desensitization) [16]. The other process, perhaps even more important, operates on the active G protein. The main mechanism for G protein signaling termination is GTP hydrolysis to GDP by the intrinsic GTPase activity of the Gα subunit [12]. As soon as GTP is converted to GDP, the GDP-bound Gα subunit regains its affinity for the Gβγ subunits (the switch II region loses its contacts with the guanine nucleotide and binds Gβ again), and the G protein heterotrimer re-associates, no longer being able to transduce signals (i.e., neither Gα nor Gβγ can interact with effectors now) [12]. 

Unlike the monomeric Ras-like G proteins, which lack intrinsic GTPase activity and depend on separate proteins that act as GTPase activating proteins (GAPs) to hydrolyze their bound GTP, the Gα subunits of heterotrimeric G proteins do possess intrinsic GTPase activity [12]. Thanks mainly to two highly conserved residues, an arginine in the switch I region and a glutamine in the switch II region, Gα subunits execute a Mg^2+^-assisted hydrolysis of GTP to GDP and inorganic phosphate [12]. Ras-like G proteins lack that conserved arginine, and this is why they lack intrinsic GTPase activity (Ras-GAPs provide that important arginine for the catalysis of GTP hydrolysis) [12]. That conserved arginine of the Gαs subunit is also the substrate that obtains ADP (adenosine diphosphate) ribosylated by cholera toxin, rendering Gαs incapable of hydrolyzing GTP and thus locking it in a permanently active conformation [17]. The rates of GTP hydrolysis vary considerably for the various Gα subunits, with certain isoforms (Gαq and Gαz) being extremely slow at converting GTP to GDP [12,18,19]. Even for the fastest GTP-hydrolyzing Gα subunits, though, their hydrolysis rates in vitro appear extremely slow and would probably be incompatible with in vivo G protein function, which usually requires very rapid termination (especially when the G protein regulates ion channel opening). Indeed, several observations in the early to mid-1990s led to the realization that the termination of G protein signals was vastly (about a hundred times) faster in various tissues in vivo than predicted by the GTP hydrolysis rate calculated in vitro [20,21,22]. These findings suggested that there were additional mechanisms/mediators involved in G protein inactivation inside living cells and, indeed, led to the discovery of the “regulator of G protein signaling (RGS)” domain, a ~120-amino-acid-long domain that can bind the Gα subunit and dramatically accelerate GTP hydrolysis [20,21,22,23,24,25,26,27,28,29,30]. The proteins that contain this RGS domain are called RGS proteins. The biological function of the RGS domain is to stabilize the GTP hydrolysis transition state of the Gα subunit (the pentavalent transition state coordinated by a Mg^2+^ cation that primes the γ-phosphate for the in-line nucleophilic attack by a water molecule), lowering the free energy barrier that needs to be overcome for the Gα subunit to carry out the hydrolytic reaction [12,30]. In other words, they act as GAPs for the Gα subunits of heterotrimeric G proteins, although, unlike their counterparts that act on monomeric G proteins, they do not participate with any actual amino acids of their own in the GTP catalysis [12]. GTP hydrolysis is enormously (up to 2000 times higher) accelerated by RGS proteins, and both the amplitude and duration of Gα and free Gβγ subunit signaling are markedly reduced [12,31]. To date, about 37 RGS proteins are known to exist, with many more identified as containing a non-functional “RGS homology” domain [26,27]. Every protein with a functional RGS domain is categorized into a subfamily, designated by a letter (A-F) and the name of a representative member of that particular subfamily (next to the letter “R”) [31,32,33,34,35]. For instance, the A/RZ subfamily is named after the representative RGSZ protein member [36]. Each subfamily grouping is based on amino acid sequence homology, structure, and function. Members of the C/R7 family, for example, have a unique DEP (disheveled, EGL-10, pleckstrin) domain, known also as the R7H domain [26,27]. The A/RZ and B/R4 subfamilies comprise members that contain essentially nothing more than just the functional RGS domain [26,27]. Of note, in addition to accelerating Gα inactivation, some RGS proteins, such as RGS4 and RGS2, can also interfere with the interaction of active (GTP-bound) Gα subunits with downstream effectors, i.e., with the activation of enzymes and other effectors by Gα per se [26]. On the other hand, by accelerating GTP hydrolysis on Gα subunits, which leads to re-association with the signaling-competent free Gβγ subunits, RGS proteins also accelerate free Gβγ subunit signaling termination [26,27]. It was initially thought that there might be a specific RGS protein for each of the >16 different mammalian Gα subunits (exactly 16 in humans), but we now know that this could not have been further from the truth [27]. Not only do the RGS proteins outnumber the Gα subunits, but also, several of them can act upon more than one Gα type/family (e.g., RGS4 inactivates both Gα_i/o_ and Gα_q/11_ subunits). However, it seems that most (if not all) RGS proteins inactivate G proteins in a cell type- and GPCR-specific manner, i.e., they do not inactivate their Gα subunit substrates at all times or under any circumstances [27]. The identity of the receptor that has stimulated the G protein seems to play a crucial role in whether that G protein serves as a substrate for the RGS protein. For example, RGS3 inactivates M_3_ muscarinic receptor- and gonadotropin-releasing hormone (GnRH) receptor-stimulated Gαq but not angiotensin II type 1 receptor (AT_1_R)-stimulated Gαq [37,38]. This is extremely important to consider because it bestows RGS protein functions with exceptional receptor-G protein signaling pathway specificity that can be exploited for therapeutic purposes. 

In the present review, we first discuss the current literature on the regulation of cardiac GPCRs by RGS proteins in the context of heart physiology but also of heart disease, followed by a closer look at cardiac RGS4, which has been documented to be implicated in human HF and atrial fibrillation (AFib). Our review focuses exclusively on the B/R4 family of RGS proteins (RGS1-5, RGS8, RGS13, RGS16, RGS18, and RGS21), the smallest mammalian RGS protein family members that function primarily (if not exclusively) as G protein GAPs, i.e., are bona fide RGS proteins. Other proteins that contain RGS homology domains but serve other primary functions (e.g., GRKs, which are serine/threonine kinases) are beyond the scope of the present review. 

## 2. RGS Proteins and GPCR Signaling in the Heart

RGS1, RGS2, and RGS3 are expressed in both cardiac myocytes and fibroblasts in vivo. RGS2 is also robustly expressed in both vascular smooth muscle and endothelial cells [32,39]. RGS4 displays the highest level of expression in the brain and in the heart, with significant expression in adrenal glands as well [24,30,31]. RGS5 is present at high levels in the vasculature, including micro-vessels (capillaries and arterioles), the aorta, and the carotid artery [34,40,41]. RGS8, RGS13, and RGS18 are mainly expressed in immune cells (B and T lymphocytes, natural killer (NK) cells, and bone marrow progenitor cells), but RGS18 has been reported in platelets, while RGS16 and RGS21 have been reported in the heart [34,42,43,44]. RGS3, the largest of the canonical B/R4 RGS proteins, exists in multiple isoforms [34]. The PDZ-containing isoform of RGS3 is expressed in cardiac atria, whereas the long and short isoforms of RGS3 are more abundant in the ventricles [45]. All the different types of cardiac cells, coronary endothelial, coronary smooth muscle, cardiomyocytes, and cardiac fibroblasts seem to express RGS3 [34,45]. In human aortic smooth muscle cells, RGS3 regulates sphingosine 1-phosphate receptor (S_1_PR), endothelin-1 (ET-1) receptor (ETR), and AT_1_R signaling [45]. Importantly, the cardiac overexpression of RGS3 inhibits maladaptive hypertrophy and fibrosis and improves cardiac function by blocking mitogen-activated protein kinase (MAPK)/extracellular-signal-regulated kinase (ERK)-kinase (MEK)-ERK1/2 signaling in transgenic mouse hearts [46] (Table 1). In that study, mice overexpressing human RGS3 specifically in the heart and developing cardiac hypertrophy secondary to aortic banding had markedly reduced hypertrophy, fibrosis, overall adverse remodeling, and better left ventricular function compared to their wild-type counterparts, in response to aortic banding [46]. These beneficial effects were attributed to cardiac AT_1_R-dependent MEK-ERK inhibition by RGS3. Cardiac RGS3 is also upregulated in spontaneously hypertensive HF (SHHF) rats [47]. However, an SHHF rat model developed congestive HF features and RGS3 mRNA and protein downregulation in the chronically failing myocardium [47]. Consistent with these findings obtained in animal models, RGS3 mRNA and protein were significantly elevated in myocardial samples from human end-stage HF patients, suggesting a role for RGS3 in human chronic and advanced HF [48]. Thus, cardiac RGS3 may play important roles in the modulation of cardiac hypertrophy and HF progression, as well as in the regulation of cardiac function in general. The specific GPCRs involved in the effects of cardiac RGS3, and also whether the association of cardiac RGS3 expression changes with human HF is causative or circumstantial, await elucidation in future studies. 

Cardiac RGS4 is most abundant in the sinoatrial (SA) and atrioventricular (AV) nodal regions, as well as throughout the atria [49,50]. It is also expressed in aorta and in ventricles [45,47,48]. Its functions in the heart are discussed in detail in the following sections below. RGS2 plays a critical role in vascular tone regulation but has been shown to affect cardiac compensation to pressure overload and to mediate the anti-hypertrophic and cardioprotective cyclic 3′, 5′-guanosine monophosphate (cGMP)-dependent effects of the phosphodiesterase (PDE)-5 selective inhibitor drug sildenafil [51]. It also appears to be involved in the counter-regulatory effects of atrial natriuretic factor (ANF) against AT_1_R-induced hypertrophy, which are mediated by the atrial natriuretic peptide receptor NPR1 [52]. NPR1 is a membrane receptor with intrinsic guanylyl cyclase activity (not a GPCR), synthesizing the second messenger cGMP that activates protein kinase G (PKG) [52]. Notably, RGS2 is the only RGS protein reported to date to directly oppose Gs protein signaling, albeit not by acting as a GAP for Gαs but rather by interacting with adenylyl cyclase (the effector of Gαs) and inhibiting it [53,54,55]. No RGS protein acting as Gαs-GAP has been reported to date [36]. RGS5 has also been reported to participate in cardio-protection against pressure overload via the inhibition of MAPK/ERK-mediated signaling [56]. However, no specific GPCR or other type of receptor was examined in that study. RGS5 or RGS2 knockouts lead to worsened pressure overload-induced cardiac fibrosis in mice [51,56]. The G_q/11_-coupled receptors AT_1_R and ET_A_R are major profibrotic mediators in human cardiac fibroblasts in response to angiotensin II and endothelin, respectively [57,58,59]. RGS2 is known to oppose AT_1_R signaling-dependent cell proliferation and collagen synthesis in ventricular fibroblasts [60]. Given that RGS2 is also expressed in cardiomyocytes, however, it is difficult to ascertain whether its anti-fibrotic actions are exerted in cardiac fibroblasts or mediated by cardiac myocytes affecting fibroblasts in a paracrine manner. The answer is probably both, but a definitive one can only come from animal models with fibroblast-restricted RGS2 deletion.

RGS13 is one of the two RGS proteins (the other one being RGS2) that typically localize in the cell nucleus [61]. Indeed, upon cyclic 3′, 5′-adenosine monophosphate (cAMP) synthesis and cAMP-dependent protein kinase (PKA) activation, RGS13 translocates to the nucleus and interacts with the PKA-phosphorylated transcription factor cAMP response element-binding (CREB) protein, inhibiting gene transcription downstream of CREB [62]. Whether this occurs in cardiac cells, however, remains an open question, given that RGS13 expression in the heart is very low (and mostly observed in cardiac fibroblasts) [31]. RGS16 is present in both cardiac myocytes and fibroblasts [31,63,64] and is one of the very few B/R4 family members known to date that act as Gα_12/13_-GAPs [65]. Bacterial lipopolysaccharide (LPS) endotoxin is associated with the impairment of myocardial contractility and acute septic HF [66]. The treatment of cultured rat cardiomyocytes with LPS, but also with ET-1 activating ET_B_Rs, upregulates RGS16 transcriptionally, dampening, in turn, phospholipase C (PLC)-β activation by ET-1-activated ET_A_Rs in cardiac myocytes [32,67]. 

## 3. Role of Cardiac RGS4 in Human HF

The exogenous overexpression of RGS4 in cardiomyocytes attenuates the G_q/11_ signaling of endothelin type A receptors (ET_A_Rs), reducing PLCβ activation and cardiomyocyte contractility but also hypertrophy [68,69,70]. Indeed, RGS4 overexpression in murine cardiac myocytes inhibits the ability of the heart to compensate for an acute increase in afterload induced by aortic banding [69]. In a study published more than 20 years ago, notably, the very first one in transgenic animals with genetically manipulated RGS protein-encoding genes, cardiac RGS4-overexpressing mice exhibited a marked increase in postoperative mortality following tight or loose aortic banding [69]. This would be expected to occur due to the reduced Gq-signaling-dependent capacity of the RGS4 overexpressors to sustain an adaptive hypertrophic/inotropic response to the increased aortic-banding-induced afterload. After all, RGS4 is known to terminate/oppose the G_q/11_-mediated signaling of ETRs, AT_1_Rs, and α_1_-adrenergic receptors (ARs) activated by phenylephrine in cardiomyocytes, which is critical for cardiomyocyte growth and cardiac cellular hypertrophy, such as that induced by increased afterload [68]. Surprisingly, however, the positive inotropic response to the βAR catecholamine agonist dobutamine was preserved in the RGS4-overexpressing mice [69], so no impairment in β-adrenergic-dependent contractility was observed that could play a role in the acute mortality post-aortic banding. The culprit for the relative inability of RGS4-overexpressing mice to acutely survive the aortic banding procedure might have been the G_i/o_ protein signaling inhibition that RGS4 also exerts in the heart, which is known to elicit downstream anti-apoptotic signals in the myocardium [71,72]. Importantly, that study demonstrated that, in the survivors of the aortic banding procedure, RGS4 overexpression ameliorated cardiac hypertrophy induced by pressure overload (increased afterload) and blocked the induction of the cardiac “fetal” gene program by directly opposing the Gq-protein-dependent signaling in the mouse heart in vivo [69]. These findings were subsequently confirmed by the same group of investigators in dual transgenic mice overexpressing both RGS4 and Gαq simultaneously in the same hearts [70]. Indeed, cardiac function and dimensions/structures were normalized in these dual transgenic mice by the age of 4 weeks, whereas the control Gαq (only) overexpressing mice displayed marked cardiac hypertrophy, embryonic gene expression, and depressed cardiac contractility by that age [70]. Nevertheless, the dual transgenic mice eventually developed reduced cardiac contractility by 9 weeks of age [70]. Taken together, these old studies were the first ones to establish a crucial role for RGS4 as a Gq protein signaling terminator in the heart in vivo, which can be cardio-protective against hypertrophic signals and increased afterload (e.g., hypertension), at least early in the course of a disease or after a cardiac insult. RGS4 was also upregulated in experimental rats of cardiac hypertrophy, including primary cardiomyocytes in culture in vitro and pulmonary artery-banded mice in vivo [47]. 

Importantly, RGS4 has been reported in two independent studies on different HF patient populations, one in Germany and another one in England, to be upregulated in advanced human HF [48,73]. In the German study, RGS4 was found to be selectively upregulated, i.e., the only one out of ten RGS proteins examined, at both the mRNA and protein levels, in human dilated or ischemic cardiomyopathy-related end stage HF [73]. In the English study, RGS4 mRNA and protein levels were increased in both end-stage and acute human HF [48]. Furthermore, RGS4 was found to blunt PLC activity in human left ventricular membranes via the obstruction of the pro-contractile and pro-hypertrophic Gq/PLC/Ca^2+^ signaling of ET_A_Rs [73]. Thus, the findings on cardiac RGS4 from all the animal model studies seem to be confirmed in humans, and a consensus role for cardiac RGS4 as being cardio-protective has been increasingly emerging. It is quite plausible that RGS4 upregulation serves as a compensatory mechanism of the human failing/ischemic myocardium to protect itself against the hypertrophic, maladaptive, and pro-contractile (oxygen-demand-increasing) Gq/PLC/Ca^2+^ signaling of certain GPCRs. 

In the same vein, we recently uncovered that RGS4 also opposes the G_i/o_ protein signaling of the short-chain free fatty acid receptor (FFAR)-3 in cultured cardiomyocytes [74]. FFAR3 is activated mainly by gut microbial metabolites propionate and butyrate, but also by other free fatty acids with a shorter than six-carbon-atom-long chain [75,76]. Like the other three human FFARs (FFAR1, FFAR2, and FFAR4), FFAR3 is a G_i/o_-coupled GPCR that promotes inflammation through p38 MAPK activation and interleukin (IL)-6 and IL-1β induction, fibrosis through transforming growth factor (TGF)-β induction, but also norepinephrine release (which increases sympathetic neuronal activity) via G_i/o_-derived free Gβγ-activated PLCβ activation and subsequent Ca^2+^ signaling [77,78,79,80]. RGS4 was found to be essential for the blockade of cardiac FFAR3-mediated inflammation and fibrosis, as well as for neuronal FFAR3-dependent sympatholysis that preserved cardiac βAR reserve (cardiomyocyte βAR membrane density) [74]. These findings suggest a protective role for cardiac RGS4 in reverse remodeling and in the mitigation of sympathetic nervous system hyperactivity induced by gut-microbiota-derived nutrient metabolites, such as propionic and butyric acids [74,78]. Of note, ketone bodies such as β-hydroxybutyrate have been reported to antagonize FFAR3 [77,81], so it appears that RGS4 can mimic (at least some of) the beneficial actions of ketone bodies in the heart.

Another signaling mechanism that could potentially endow RGS4 with therapeutic benefit potential in human HF is the positive regulation of cardiac cAMP levels it may exert courtesy of its GAP activity at Gαi subunits (Figure 1). As has been suggested for RGS4 in pancreatic beta cells and other tissues [82,83], the termination of Gαi subunit signaling by RGS4 would relieve adenylyl cyclase from Gαi inhibition, thereby indirectly promoting cAMP synthesis (and PKA activation) by Gs-coupled GPCRs, such as the cardiac βARs (Figure 1). The fact that the response of the RGS4-overexpressing mice to dobutamine post-aortic banding was normal also argues in favor of this scenario [69]. This mechanism might be particularly important in the setting of human HF, given that Gαi (but not Gαs or Gαq) is known to be selectively upregulated in the failing human heart, regardless of the type of failure (acute or chronic end-stage) or etiology (ischemic or dilated cardiomyopathy) [84,85,86,87,88,89,90] (Figure 1). This Gαi upregulation is driven by the norepinephrine overstimulation of cardiac β_1_ARs, which transcriptionally upregulate Gαi via the Gs protein/cAMP/PKA signaling axis, and thus probably serves as a feedback, counter-regulatory mechanism against the catecholaminergic overdrive of the failing heart [86,87]. However, increased Gαi activity means that basal and hormone-activated adenylyl cyclase activities are suppressed, leading to chronically low cAMP levels in the failing human heart (Figure 1). Indeed, several lines of evidence point to the fact that cAMP levels are low and cAMP synthesis is deficient in the failing human heart [91,92,93,94]. Although this might initially serve as an adaptive response of the failing myocardium to protect itself from excessive norepinephrine stimulation (the developing sympathetic nervous system overdrive), low cAMP levels can become maladaptive over time, because cAMP is essential not only for the contractile (systolic) function of the heart but also for its relaxation (diastolic) function [84,88]. In addition to inotropy, automaticity, and dromotropy, cAMP increases the lusitropy of the myocardium, as well. This is mainly achieved by a combination of PKA-dependent phosphorylations that primarily activate SERCA in the sarcoplasmic reticulum (SR) (via phospholamban phosphorylation) to remove Ca^2+^ from the cytoplasm back into the SR [95], the sodium pump in the plasma membrane (via phospholemman phosphorylation) to drive sodium/calcium exchanger (NCX)-mediated Ca^2+^ extrusion out of the cardiomyocyte [96], and even accelerate actomyosin filament relaxation (via myosin-binding protein-C3, MyBPC3, and phosphorylation) [97,98] (Figure 1). All these actions combined reverse the intracellular free [Ca^2+^] elevation induced by cAMP during contraction and allow for the myocardium to relax and fill with blood during diastole [99,100]. It is thus quite plausible that RGS4 is selectively (among all RGS proteins expressed in the human heart) upregulated in the failing human heart as a compensatory mechanism for the myocardium in an effort to counterbalance the Gαi upregulation and maintain some basic level of adenylyl cyclase activity and cAMP synthesis necessary for proper cardiomyocyte homeostasis (Figure 1). In fact, one of the first articles reporting the Gαi upregulation in human HF, by Bohm and colleagues in 1990, ends with the quote: “Inactivation of Giα could be a potential target for the medical treatment of chronic heart failure” [85]. The RGS proteins discovered a few years later, specifically RGS4, could fill this role perfectly. Nevertheless, this RGS4 upregulation is evidently insufficient to increase cAMP levels in the failing human heart to a substantial extent, given that the cAMP levels measured in advanced human HF are still low [91,92]. Thus, RGS4 upregulation alone does not suffice to halt (let alone reverse) the progressive deterioration of cardiac function in human chronic HF. Interestingly, cardiac RGS4 has been documented to be activated, i.e., its plasma membrane recruitment to be stimulated, by PKA-dependent phosphorylation, which is, in turn, induced by βAR activation [74]. Therefore, the low cAMP levels that accompany human HF may not be conducive to RGS4 activation reaching sufficient levels to counteract the Gαi upregulation of the failing human heart. Finally, it is worth noting that the fact that Gαi is elevated in the failing human heart means that interventions such as GRK2 inhibition, aimed at increasing βAR-elicited Gs protein signaling that is depressed in human HF due to elevated GRK2-dependent desensitization [101,102], would probably be ineffective at sufficiently improving cAMP levels and, consequently, cardiac function.

## 4. Role of Cardiac RGS4 in Human AFib

Apart from its putative roles in the regulation of cardiac inotropy and lusitropy, RGS4 has been documented to play a crucial role in cardiac chronotropy regulation [103]. The cholinergic regulation of heart rate is mainly mediated by the G_i/o_-coupled M_2_ muscarinic cholinergic receptor (mAChR) expressed throughout the human atria and in the pacemaker regions (SA and AV nodes) [91,104]. The main signaling pathway underlying the acetylcholine (ACh)-induced slowing of cardiac conduction (bradycardia) involves the activation of G_i/o_-derived free Gβγ subunits, which, in turn, activate atrial G protein-coupled inwardly rectifying K^+^ (GIRK) channels, resulting in acetylcholine (ACh)-induced K^+^ hyperpolarizing currents (IKACh) [27,103,104]. The M_2_ mAChR-stimulated, as well as adenosine receptor-stimulated Gαi-dependent inhibition of adenylyl cyclase also contributes to the cholinergic (and adenosinergic) slowing of heart rate (HR), since cAMP is essential for the operation of Hyperpolarization-activated Cyclic Nucleotide-gated (HCN)-4 channels, responsible for the generation of the pacemaker “funny” current (If) in SA nodal pacemaker cells [105,106] (Figure 1). cAMP also enhances depolarizing Ca^2+^ influx currents in AV nodal cells (via the PKA-mediated phosphorylation and opening of L-type calcium channels (LTCCs) and of ryanodine receptor (RyR2) channels), which is responsible for the propagation of electrical conduction throughout the atria, AV node, and over to the ventricles (Purkinje fibers and Hiss bundle) [26,27,95,107] (Figure 1). In other words, cAMP lowering reduces automaticity and induces negative dromotropy in the heart. RGS4 and RGS6 have long been known to function as key regulators of the cholinergic control of HR [108,109,110]. The knockout of either RGS4 or RGS6 produces phenotypes of severe bradycardia and AV/heart block in mice in response to vagal stimulation in vivo [108,109,110]. The underlying mechanisms of this negative chronotropy regulation may not be fully shared between RGS4 and RGS6, given that RGS6, but not RGS4, can directly interact with Gβ5 via its Gγ-like (GGL) domain and form a complex that inactivates the ACh-activated G protein-gated K^+^ channel (IKACh) [108].

In fact, the role of RGS4 in the negative regulation of normal, basal IKACh currents in the SA node has been challenged by several studies [111,112]. Indeed, it appears that, under normal, basal vagal tone conditions, RGS6 and RGS10 are mainly responsible for IKACh desensitization [111,113]. Upon conditions that enhance vagal tone, however, such as intense chronic exercise in humans, RGS4 takes over and suppresses the excess IKACh currents that promote AFib development secondary to physical exercise or other AFib-precipitating stimuli [110,112]. Indeed, athletes, particularly endurance athletes (marathon runners, etc.), are at increased risk of developing AFib due to autonomic imbalances of HR regulation [112]. Animal models of exercise-induced AFib exhibit atrial fibrosis, dilation, and heightened vagal tone with significantly increased IKACh currents, but, interestingly, their sympathetic tone is unchanged [112]. Importantly, the cause of the elevated IKACh currents, predisposing one to AFib via increased atrial refractoriness, was found to be neither changes in adrenergic or muscarinic receptor expression, nor changes in the G protein-gated inwardly rectifying K^+^ channel (GIRK) subunits themselves or in the G proteins that help activate them (phosphatidylinositol 4′, 5′-bisphosphate, PIP_2_, actually serves as the natural agonist of GIRKs), i.e., Gi proteins [112]. Instead, the downregulation of several RGS proteins (RGS4 included) was found to be the culprit, while, interestingly, RGS9, which, unlike the other RGS proteins, actually promotes IKACh, was upregulated [112]. Of course, the caveat of these results was that expression changes were measured at the mRNA level only (not protein). Nevertheless, the authors of that study went on to confirm the crucial role (specifically) of RGS4 in protection against exercise-induced vagal tone enhancement and subsequent AFib pathogenesis, since RGS4 knockout mice were found extremely vulnerable to AFib development upon cholinergic stimulation, i.e., were much more susceptible to carbachol-stimulated AFib than wild-type controls [112]. 

Further supporting a cardioprotective role for RGS4 against AFib pathogenesis, in addition to its role in IKACh regulation, is the fact that RGS4 is essential for the suppression of pro-arrhythmogenic Ca^2+^ signaling by G_q/11_ protein-coupled receptors, primarily the endothelin ET_A_ and angiotensin II AT_1_ receptors, in the heart [114]. Indeed, RGS4 knockout mice developed atrial-burst-pacing-induced AFib more frequently than control wild-type littermates [114]. Isolated atrial cells lacking RGS4 displayed higher Ca^2+^ spark frequencies both under basal conditions and upon ET-1 treatment [114]. Abnormal Ca^2+^ release events were also more often observed in RGS4 knockout myocytes [114]. Thus, RGS4 deletion predisposes one to AFib due to elevated/unchecked G_q/11_-PLCβ/inositol trisphosphate (IP_3_)/Ca^2+^ signaling resulting in abnormal, ectopic beats/electrical events [114]. Moreover, RGS4 has been shown to suppress PLC activity (and subsequent Ca^2+^ signaling), both basally and upon ET-1 stimulation, in human cardiomyocyte membranes [73]. Finally, the NLRP3 (nucleotide-binding domain, leucine-rich-containing family, pyrin domain-containing-3) inflammasome is known to play pivotal roles in AFib pathogenesis [115,116]. cAMP inhibits NLRP3 via PKA-dependent phosphorylation, so Gαs-coupled receptors can block, whereas Gαi can stimulate NLPR3 inflammasome activity [117]. Gq/Ca^2+^ signaling, e.g., from adenosine triphosphate (ATP) purinergic or calcium-sensing receptors, as well as adipokines and short-chain fatty acid metabolites secreted by epicardial fat adipocytes, also potently activate the NLRP3 inflammasome and exacerbate AFib [115,117]. It is thus quite plausible that RGS4, through its opposing actions on Gi and Gq protein signaling, can also suppress NLRP3 inflammasome activation in both atrial myocytes and epicardial adipocytes [118] (Figure 2). This could constitute another mechanism by which RGS4 protects the myocardium against AFib development. In conclusion, RGS4 appears to be essential for the suppression of excessive Ca^2+^ and cholinergic IKACh signaling in human atria, both of which are arrhythmogenic and can lead to AFib development (Figure 2). This strongly suggests that pharmacological interventions to enhance cardiac RGS4 expression and/or activity might have significant therapeutic value in AFib treatment and prevention, especially since RGS4 does not seem to negatively affect normal vagal HR regulation, which would be arrhythmogenic on its own and also appears to be protective against pathological cardiac hypertrophy [119].

## 5. Conclusions and Future Perspectives

Considerable progress has been made over the past two decades in delineating the various signaling properties and biological actions of RGS proteins in almost every organ system, including in the cardiovascular system. As key regulators of GPCR signaling through G proteins, RGS proteins are enticing therapeutic targets based on their physiological and pathophysiological importance in the heart, kidneys, central nervous system, oncology, and other disease areas. Either RGS protein inhibition or potentiation can be beneficial therapeutically, depending on the individual receptor/G protein pathway/tissue setting in question. Of course, RGS protein inhibition should always theoretically lead to enhanced GPCR signaling, which can be advantageous in that it can allow for a reduced dosage (and hence, side effects) of the GPCR agonist given as a drug (e.g., increased β_2_AR signaling in bronchial smooth muscle in asthma or increased M_2_ mAChR signaling in tachycardias). Additionally, by blocking the activation of certain effectors by certain G proteins (e.g., the RGS2-mediated blockade of adenylyl cyclase activation by Gαs and the RGS4-mediated blockade of PLCβ by G_i/o_-derived free Gβγ), RGS protein inhibitors can fine-tune the specificity of GPCR signaling in response to GPCR agonists administered as drugs. On the other hand, there is a plethora of pathological situations in which enhanced G protein signaling, accompanied by reduced RGS protein activity, is involved in the pathophysiology of a cardiovascular disease, meaning the augmentation of RGS protein function would be desirable. 

Although a considerable amount of work still needs to be done to fully elucidate its function in the heart and in other organs, RGS4 has already emerged as a potential therapeutic target in both human AFib and HF. Coupled with its potential in the treatment of kidney injury/disease [120], cancer [121,122], asthma [123], and diabetes [83], the development of a pharmacological “magic bullet” based on RGS4 activity augmentation in the future will not be surprising. Hopefully, the advent and continuing development of isoform-specific, reversible, and potent small organic molecules (or cell-permeable peptides) will allow for the complete evaluation of the potential of RGS protein pharmacological targeting to accurately define their ultimate place in the list of drug discovery targets for the treatment of cardiac hypertrophy, HF, AFib, arrhythmias, hypertension, and many other heart diseases.

## Figures and Tables

**Figure 1 ijms-24-06136-f001:**
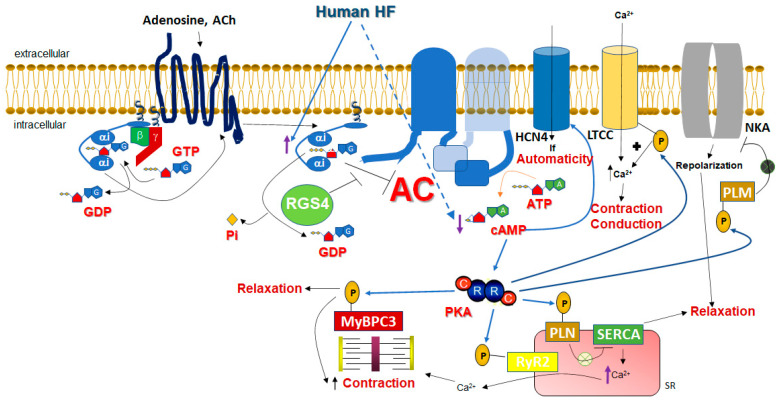
Role of cardiomyocyte RGS4 in the context of human HF. Gαi activity is high and cAMP levels are low in the failing human heart. Phosphorylation of MyBPC3 by PKA first increases contraction before it facilitates relaxation of the actin-myosin fibers. ACh: acetylcholine; AC: adenylyl cyclase; Pi: inorganic phosphate; P: phosphorylation; NKA: Na^+^/K^+^-adenosine triphosphatase (sodium pump); PLM: phospholemman; PLN: phospholamban; ATP: adenosine triphosphate; SR: sarcoplasmic reticulum; SERCA: sarco(endo)plasmic reticulum calcium adenosine triphosphatase; A: adenine; G: guanine; R: regulatory subunit of PKA; C: catalytic subunit of PKA. See text for details and all other molecular acronym descriptions.

**Figure 2 ijms-24-06136-f002:**
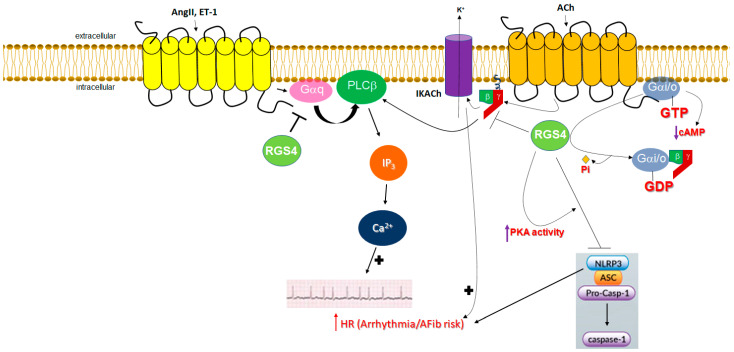
Role of (atrial) cardiomyocyte RGS4 in the context of human AFib. Note that the free Gi/o-protein-derived Gβγ dimer can also activate certain isoforms of PLCβ directly. Also depicted is the inhibitory effect of RGS4 on NLRP3 inflammasome (indirectly, via enhanced PKA-dependent phosphorylation) which awaits experimental confirmation. AngII: angiotensin II; ACh: acetylcholine; IP_3_: inositol 1′, 4′, 5′-trisphosphate; Pi: inorganic phosphate; ASC: apoptosis-associated speck-like protein containing a caspase recruitment domain (CARD); Pro-Casp-1: pro-caspase-1; HR: heart rate. See text for details and all other molecular acronym descriptions.

**Table 1 ijms-24-06136-t001:** Summary of the most important documented effects of cardiac RGS proteins.

RGS Isoform	Effects in the Heart
RGS3	↓ Cardiac hypertrophy/remodeling in response to PO; ↑ Cardiac function in response to PO
RGS4	↓ Cardiac hypertrophy/remodeling in response to PO; ↓ Cardiac arrhythmogenesis/AFib risk; ↓ Cardiac inflammation/adverse remodeling; ↓ NE release from SNS neurons
RGS16	↑ Cardio-protection against LPS/sepsis
RGS2; RGS4; RGS6	HR regulation

PO: pressure overload; LPS: lipopolysaccharide; NE: norepinephrine; SNS: sympathetic nervous system; HR: heart rate; AFib: atrial fibrillation.

## Data Availability

Not applicable.

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
