# Peer review of "Cardiac RGS Proteins in Human Heart Failure and Atrial Fibrillation: Focus on RGS4"

_ijms, 2023, doi:10.3390/ijms24076136_

Round 1
Reviewer 1 Report
Borges et al. provide a thoughtful summary of the current literature surrounding RGS regulation of GPCR signalling with a focus in cardiovascular disease. The authors provide an appropriately-detailed background about GPCR and RGS signalling. This is followed by delving into the role of RGS4 in HF and AFib. This review is well-written and informative. I have a few minor comments to help improve the overall manuscript.
1. Many RGS proteins are discussed in section 2 and can get a bit confusing for the reader. A table summarizing what is known about different RGS proteins in the heart would be helpful.
2. In line 155 is it meant to say ‘implicated in human heart failure and atrial fibrillation’?
3. RGS4 appears to play an important role in terminating G
i inhibition of adenylyl cyclase in HF, however the authors should address how this also impacts the downstream effects of G
signalling, if known or comment if it is not clear.
4. I recommend citing the original study showing cardiomyocyte NLRP3 activation is involved with AFib pathogenesis and progression.
doi: 10.1161/CIRCULATIONAHA.118.035202
5. The potential role of G
s coupled receptors in supressing NLRP3 activity via cAMP/PKA activation is interesting. I recommend including a visualization of this in Figure 2.
6. Have there been any randomised-controlled clinical trials (or any clinical phase studies whatsoever) assessing RGS-modifying drugs in cardiovascular disease or any pathology for that matter?
Author Response
Borges et al. provide a thoughtful summary of the current literature surrounding RGS regulation of GPCR signalling with a focus in cardiovascular disease. The authors provide an appropriately-detailed background about GPCR and RGS signalling. This is followed by delving into the role of RGS4 in HF and AFib. This review is well-written and informative. I have a few minor comments to help improve the overall manuscript.
- Many RGS proteins are discussed in section 2 and can get a bit confusing for the reader. A table summarizing what is known about different RGS proteins in the heart would be helpful.
Author response: We thank this reviewer for the overall kind and positive comments about the quality of our work. The requested table has been added (Table 1 of the revised manuscript).
- In line 155 is it meant to say ‘implicated in human heart failureand atrial fibrillation’?
Author response: Yes, thank you. This typo has been corrected.
- RGS4 appears to play an important role in terminating Gi inhibition of adenylyl cyclase in HF, however the authors should address how this also impacts the downstream effects of Gsignalling, if known or comment if it is not clear.
Author response: We thank the reviewer for this pertinent remark. We are not sure what he/she means here exactly but we believe we have covered all the physiologically relevant downstream effects of cardiac-specific G protein signaling that RGS4 is known to affect.
- I recommend citing the original study showing cardiomyocyte NLRP3 activation is involved with AFib pathogenesis and progression.
doi: 10.1161/CIRCULATIONAHA.118.035202
Author response: Done. Thank you for this excellent recommendation!
- The potential role of Gs coupled receptors in supressing NLRP3 activity via cAMP/PKA activation is interesting. I recommend including a visualization of this in Figure 2.
Author response: We thank this reviewer for another excellent suggestion. We have modified Figure 2 accordingly.
- Have there been any randomised-controlled clinical trials (or any clinical phase studies whatsoever) assessing RGS-modifying drugs in cardiovascular disease or any pathology for that matter?
Author response: This is an interesting question raised by this reviewer. The answer is, to our knowledge, negative, i.e., not yet. There are two main reasons for this: a) The function of any given RGS protein isoform in any given organ/tissue needs to be fully delineated first, in order to ascertain whether augmentation or inhibition of that RGS protein`s function is therapeutically desirable, and b) No drugs fully specific for any particular RGS protein have been developed yet.
Reviewer 2 Report
The manuscript is of interest; however, there are several deficits needing to be improved for the publication of IJMS.
1. There are several typos in the current form of this manuscript, please carefully revise them.
2. The definition of Ach in the legend of figure 1 needs to be provided.
3. The style of references was not consistent throughout the reference section, please revised.
4. Please use heart failure or HF uniformly in this manuscript.
5. Can norepinephrine or epinephrin induce the activation of RGSs? Please discuss it in the revised manuscript.
Author Response
The manuscript is of interest; however, there are several deficits needing to be improved for the publication of IJMS.
- There are several typos in the current form of this manuscript, please carefully revise them.
Author response: We thank this reviewer for the kind and positive comments about the quality of our work. We have proofread our text and corrected all typos.
- The definition of Ach in the legend of figure 1 needs to be provided.
Author response: Done, thank you.
- The style of references was not consistent throughout the reference section, please revised.
Author response: Done, thank you.
- Please use heart failure or HF uniformly in this manuscript.
Author response: Done, thank you.
- Can norepinephrine or epinephrin induce the activation of RGSs? Please discuss it in the revised manuscript.
Author response: We thank this reviewer for this pertinent remark. Unfortunately, the answer to this question is very complicated because it depends on the specific RGS protein in question and also on which receptor the catecholamines activate. To keep it simple and, since our present review focuses on RGS4, this isoform is known to be activated by PKA phosphorylation, which means that norepinephrine and epinephrine can activate RGS4 via beta-adrenoceptors. We have added a brief discussion of this in our revised manuscript, as follows: “Interestingly, cardiac RGS4 has been documented to get activated, i.e., its plasma membrane recruitment to be stimulated, by PKA-dependent phosphorylation, which is, in turn, induced by βAR activation [74]. Therefore, the low cAMP levels that accompany human HF may not be conducive to RGS4 activation reaching sufficient levels to counteract the Gai upregulation of the failing human heart”, lines 347-352 of the revised text (also highlighted in yellow). We hope this satisfies now this reviewer.
Round 2
Reviewer 1 Report
Thank you for addressing the comments. I feel this manuscript is sufficient to publish